# Symphony of Digestion: Coordinated Host–Microbiome Enzymatic Interplay in Gut Ecosystem

**DOI:** 10.3390/biom15081151

**Published:** 2025-08-11

**Authors:** Volodymyr I. Lushchak

**Affiliations:** 1Department of Biochemistry and Biotechnology, Vasyl Stefanyk Precarpathian National University, Ivano-Frankivsk, 57 Shevchenko Str., 76018 Ivano-Frankivsk, Ukraine; volodymyr.lushchak@pnu.edu.ua; 2Research and Development University, Ivano-Frankivsk, 13a Shota Rustaveli Str., 76018 Ivano-Frankivsk, Ukraine

**Keywords:** gut microbiota, digestive enzymes, nutrient sensing, host–microbiome interaction, intestinal barrier, oxidative stress, microbial metabolism, precision nutrition, hormonal regulation, short-chain fatty acids (SCFAs)

## Abstract

Digestion was once viewed as a host-driven process, dependent on salivary, gastric, pancreatic, and intestinal enzymes to break down macronutrients. However, new insights into the gut microbiota have redefined this view, highlighting digestion as a cooperative effort between host and microbial enzymes. Host enzymes initiate nutrient breakdown, while microbial enzymes, especially in the colon, extend this process by fermenting resistant polysaccharides, modifying bile acids, and transforming phytochemicals and xenobiotics into bioactive compounds. These microbial actions produce metabolites like short-chain fatty acids, which influence gut barrier function, immune regulation, and metabolism. I propose two frameworks to describe this interaction: the “duet,” emphasizing sequential enzymatic cooperation, and the “orchestra,” reflecting a spatially and temporally coordinated system with host–microbiota feedback. Disruption of this symbiosis, through antibiotics, inflammation, diet, or aging, leads to dysbiosis, impaired digestion, and contributes to metabolic, neurologic, cardiovascular, and inflammatory diseases. Recognizing digestion as a dynamic, integrated system opens new paths for therapies and nutrition. These include enzyme-targeted prebiotics, probiotics, postbiotics, and personalized diets. Embracing this systems-level perspective enables innovative diagnostics and treatments, aiming to restore enzymatic balance and improve digestive and systemic health.

## 1. Introduction

Digestion is a fundamental biological process that enables the breakdown of complex food matrices into absorbable nutrients essential for human survival and health. Traditionally, the gastrointestinal tract (GIT) has been viewed primarily as a mechanical and enzymatic system orchestrated by the host’s secretions, including salivary amylase, gastric pepsin, pancreatic amylase, proteases, lipases, and a suite of brush-border enzymes. This classical perspective posits that human digestive enzymes dominate in nutrient processing, functioning largely independently of other biological factors. However, advances in microbial ecology, molecular biology, and high-throughput sequencing have profoundly reshaped our understanding of this intricate system. The gut microbiota—the vast and diverse community of microorganisms residing in the digestive tract—has emerged as a pivotal player in digestion, contributing a vast arsenal of enzymes capable of degrading dietary components that the host enzymes cannot efficiently process on their own [1,2,3].

The emerging paradigm recognizes the gut as a complex and dynamic ecosystem, where digestive processes result from a coordinated interplay between host-derived enzymes and microbial enzymatic activities. Rather than working in isolation or redundancy, these enzymes form a synergistic network that dynamically adapts specific genotypes to diet, health status, and environmental factors. Such complicated cooperation may be described as a musical performance, where the host and microbiota each contribute their own “instruments” to a harmonious “orchestra,” or at times engage in a more intimate “duet,” complementing and completing each other’s functions [4,5,6]. For example, human enzymes initiate the breakdown of starches and proteins, while microbial enzymes further ferment resistant carbohydrates and complex polyphenols, producing short-chain fatty acids (SCFAs) and other bioactive metabolites critical for colonic health and systemic metabolism [7,8].

Understanding this coordinated enzymatic symphony is crucial because disturbances in this relationship are linked to a range of pathologies, including inflammatory bowel disease, obesity, metabolic syndrome, and neurodegenerative disorders [9]. Moreover, the integration of host and microbial enzymatic contributions opens new avenues for therapeutic interventions, nutritional strategies, and personalized medicine aimed at restoring or optimizing digestive health. This review explores the enzymatic roles of the host and the gut microbiota in digestion, conceptual models of their cooperation, the consequences of their dysregulation, and prospects for targeting this synergy to promote health. Such a systems approach paves the way for precision nutrition, offering a powerful tool for personalized prevention and treatment of most non-communicable diseases.

## 2. The Enzymatic Roles of the Host and the Gut Microbiota

The human gastrointestinal tract operates as a highly coordinated system in which both host-derived and microbiota-derived enzymes play essential roles in the digestion and biotransformation of nutrients. Host enzymes are primarily responsible for the initial stages of macronutrient breakdown, while microbial enzymes complement and extend this process by degrading complex dietary components that escape host digestion, including dietary fibers, polyphenols, and resistant starches. This enzymatic interplay significantly influences nutrient absorption, energy harvest, immune responses, and gut barrier integrity. Table 1 summarizes the key enzymes produced by the host and gut microbiota, highlighting their substrates, products, and physiological significance.

### 2.1. Host Digestive Enzymes

The host digestive system secretes a complex array of enzymes that initiate and execute the hydrolysis of macronutrients. Digestion begins in the oral cavity with salivary amylase, which initiates starch breakdown into maltose and dextrins. The acidic environment of the stomach activates pepsinogen to pepsin, which denatures proteins and cleaves peptide bonds, facilitating further proteolysis in the small intestine [3,4,5]. The pancreas secretes a cocktail of digestive enzymes, including amylases, lipases, proteases (trypsin, chymotrypsin), and nucleases, which collectively break down carbohydrates, lipids, proteins, and nucleic acids. Finally, brush-border enzymes such as maltase, lactase, and sucrase complete the digestion of disaccharides into monosaccharides suitable for absorption [5,10].

The secretion and activity of these enzymes are tightly regulated by neural signals; hormonal controls such as gastrin, cholecystokinin, and secretin; and influenced by nutrient sensing mechanisms that modulate digestive capacity in response to diet composition and feeding status [11,12]. This adaptive regulation ensures efficient digestion under varying physiological conditions and dietary intakes. However, despite this sophisticated host enzymatic machinery, many complex polysaccharides and other dietary components remain indigested without microbial assistance.

### 2.2. Microbial Digestive Enzymes

The human gut microbiota encodes an extensive repertoire of carbohydrate-active enzymes (CAZymes) not produced by the host, including glycoside hydrolases, polysaccharide lyases, and carbohydrate esterases that degrade a wide range of complex polysaccharides such as resistant starches, cellulose, pectins, and hemicelluloses [1,7,8,13,14]. Additionally, microbial proteases complement host proteolysis by breaking down dietary and endogenous proteins in the colon, releasing peptides and amino acids that may serve as substrates for further microbial metabolism [15]. Bile salt hydrolases, another class of microbial enzymes, modulate bile acid profiles by deconjugating bile salts, thus influencing lipid digestion, cholesterol metabolism, and intestinal signaling pathways [16,17]. The microbial fermentation of undigested carbohydrates generates SCFAs—primarily acetate, propionate, and butyrate—that serve as energy sources for colonocytes, regulate immune responses, and impact systemic metabolism through signaling mechanisms [1,4,8,13,14,18]. Interestingly, microbes not only produce bioactive metabolites such as vitamins (K, B12), neurotransmitter precursors, and phenolic derivatives, highlighting their broad enzymatic influence beyond mere digestion, but also use them [19,20,21]. In addition to broadly discussed items, there are some specific routes for the transformation of specific compounds by the gut microbiota and which may substantially affect hosts. For example, gut microbiota may convert glucoraphanin, a glucosinolate found in cruciferous plants, to isothiocyanate sulforaphane and related compounds [22].

Spatially, enzymatic activities vary significantly along the gut axis. The small intestine harbors fewer microbes, predominantly facultative anaerobes, whereas the colon hosts a dense and diverse anaerobic community with extensive polysaccharide-degrading capacity. This spatial organization reflects adaptation to substrate availability and environmental conditions, further emphasizing the complementary nature of host and microbial enzymes [4,10,12].

## 3. Models of Cooperation: Duet or Orchestra?

Conceptualizing the host–microbiome enzymatic interaction is essential to understanding digestion beyond a sum of parts. Two models frame this cooperation: the “duet” and the “orchestra” (Figure 1). In the duet model (Figure 1, left), enzymatic activities are sequential or complementary—host enzymes initiate digestion, and microbial enzymes complete it, and vice versa. For instance, salivary and pancreatic amylases hydrolyze starch to maltose and oligosaccharides, which escape absorption in the small intestine and are fermented by microbial amylases and glycoside hydrolases into SCFAs in the colon [1,7,13,18,23]. Similarly, the host may degrade dietary polyphenols partially, enabling microbial enzymes to generate bioactive metabolites with systemic effects [24,25].

The orchestra model (Figure 1, right) expands this concept to a highly coordinated, multi-layered interaction where host and microbial enzymes operate in a spatially and temporally regulated manner with feedback loops. For example, bile acids secreted by the liver are modified by microbial bile salt hydrolases, which modulate bile acid signaling via nuclear receptors (FXR, TGR5), influencing digestion, host metabolism, and immune responses [17,26,27]. This multidirectional crosstalk highlights an intricate network where enzymes, microbial populations, and host cells communicate dynamically.

Metagenomics and metabolomics studies reveal that gene expression profiles of microbial enzymes change in response to diet and host physiology, underscoring the system’s plasticity [28,29]. Gnotobiotic animal models confirm that colonization by specific microbial consortia influences host enzyme expression and digestive efficiency [30,31].

## 4. Disruption of Enzymatic Harmony: Dysbiosis and Disease

The finely tuned enzymatic cooperation between host and microbiota can be disrupted by antibiotics, chronic inflammation, aging, and dietary changes, leading to dysbiosis—an imbalance in microbial composition and function [32,33,34]. Antibiotics reduce microbial diversity and enzyme pools, impairing carbohydrate fermentation and SCFA production, which negatively affects gut barrier integrity and immune homeostasis [32,35]. Inflammatory bowel diseases exhibit altered expression of microbial proteases, contributing to mucosal damage and nutrient malabsorption [36].

Aging is associated with reduced microbial diversity and enzymatic activity, contributing to diminished digestive efficiency, increased intestinal permeability, and systemic inflammation (“inflammaging”) [21,32,33,34]. Diets low in fiber reduce substrates for microbial fermentation, leading to decreased SCFA levels and altered bile acid metabolism, which are linked to metabolic disorders such as obesity and type 2 diabetes [33,37].

The loss or overexpression of specific microbial enzymes—such as excessive proteolytic activity—can produce harmful metabolites, exacerbate inflammation, and impair nutrient absorption [34,38,39]. It clearly shows the importance of enzymatic balance for maintaining gut and systemic health.

## 5. Therapeutic Perspectives and Future Directions

Targeting the enzymatic synergy in the gut opens new therapeutic opportunities. Probiotics engineered or selected for specific enzyme activities (“enzyme-targeted probiotics”) can restore deficient functions, enhance fiber degradation, or modulate bile acid metabolism [40,41]. Postbiotics—microbial metabolites or enzyme preparations—offer alternative means to supplement or regulate enzymatic activity [42,43].

Dietary modulation remains a cornerstone; prebiotic fibers selectively stimulate beneficial enzyme-expressing microbes, improving SCFA production and digestive efficiency. Personalized nutrition approaches integrating gut microbiota enzymatic profiles and host genetics promise tailored dietary recommendations for optimal digestive health.

Future research will leverage systems biology to model the complex enzymatic networks and synthetic biology to engineer microbial consortia with desired enzymatic functions. Microbiome engineering may enable the creation of designer ecosystems to prevent or treat digestive and metabolic diseases.

## 6. Some Poorly Covered or Overlooked Points

In the study of digestion, attention is mostly focused on carbohydrates, lipids, and proteins, while nucleic acids are much less covered. In the human GIT, nucleic acid digestion is a complex process involving both host and microbial enzymes that ensures the breakdown and absorption of DNA and RNA derived from diet, sloughed epithelial cells, and microbiota. Upon ingestion, nucleic acids are first denatured in the stomach by the acidic environment (pH ~1.5–3.5). It disrupts their secondary and tertiary structures, making them more accessible to enzymatic action. In the small intestine, pancreatic secretions containing deoxyribonuclease I (DNase I) and ribonuclease A (RNase A) hydrolyze DNA and RNA into smaller oligonucleotides. These are subsequently degraded by brush-border enzymes such as nucleotidases and nucleosidases into nucleosides and nitrogenous bases (e.g., adenine, guanine, cytosine, uracil, thymine), as well as pentose carbohydrates (ribose, deoxyribose). Specific nucleoside and base transporters in the enterocyte membrane, including concentrative nucleoside transporters (CNTs) and equilibrative nucleoside transporters (ENTs), facilitate their absorption into the portal circulation. However, not all nucleic acid components are absorbed in the small intestine. Residual nucleotides and nucleic acids pass into the colon, where they are metabolized by the gut microbiota. Certain microbial taxa, including *Bacteroides* spp., *Clostridium* spp., and *Faecalibacterium prausnitzii*, produce extracellular DNases and RNases, contributing to the degradation of residual nucleic acids [44,45].

The human gut microbiome harbors a diverse set of microbial species capable of expressing extracellular nucleases that degrade residual nucleic acids. These microbes utilize liberated nucleotides and bases in their purine and pyrimidine salvage pathways for DNA/RNA synthesis, or ferment nucleic acid-derived sugars into short-chain fatty acids (SCFAs) such as acetate, propionate, and butyrate—key molecules in host energy metabolism and intestinal health [41,46]. Microbial metabolism of nucleic acids also contributes to nitrogen recycling in the colon. Beyond energy metabolism, microbial nucleic acid turnover also affects host immune signaling; unmethylated CpG motifs from bacterial DNA can activate Toll-like receptor 9 (TLR9), initiating immune responses that may contribute to chronic intestinal inflammation [47,48]. Dysregulated microbial nucleic acid metabolism has been implicated in inflammatory bowel disease (IBD), where excessive CpG DNA sensing may disrupt mucosal tolerance, and moreover, imbalances in purine metabolism—potentially driven by microbial activity—have been linked to colorectal cancer through genotoxic stress and altered nucleotide pools, as well as metabolic disorders [49]. Furthermore, dietary nucleotides have been shown to influence gut development, promote beneficial microbial populations such as *Bifidobacterium*, and improve immune function in neonates and immunocompromised individuals [50]. As such, both endogenous and dietary nucleic acids, along with microbial contributions, play a crucial role in maintaining intestinal homeostasis and systemic health [51,52]. Dietary nucleotides can also modulate the gut microbiota, enhance mucosal immunity, and influence early gut development. Altogether, these observations highlight nucleic acid digestion as an overlooked but important contributor to the host–microbiota metabolic axis, with implications for immune regulation, gut barrier function, and disease pathogenesis.

Several points can be highlighted based on our previous experiments with GIT inhabitants. For example, different strains of the same bacterial species may exhibit varying sensitivity to environmental stresses. In the case of *Escherichia coli*, we observed that differences in oxygen tolerance among strains were associated with their distinct capacities to manage reactive oxygen species (ROS), particularly through the activity of antioxidant enzymes [53]. Exposure of bakery yeasts *Saccharomyces cerevisiae* to hydrogen peroxide triggered a hormetic reshaping of their phenotype [54]. Moreover, the specific pattern of this reorganization was influenced by the type of carbohydrate used during yeast cultivation. Finally, *S. cerevisiae*, under certain conditions, may produce polypeptides—enzyme precursors—which, under stress conditions, may mature to active forms [55]. Despite the above three works being carried out in in vitro conditions, they clearly show that many more parameters, as currently practiced, should be taken into consideration when dealing with such a complex GIT ecosystem.

## 7. Orchestrating Host–Gut Microbiota Cooperation: A System Conducting

The digestion of food is not a primitive, mechanical, or chemical process. It is a very complicated system where reactions carried out by the host and gut microbiota digestive enzymes are tightly regulated on short and long-term time scales. Both hormonal and neural regulations in concert govern this system. Similarly to an orchestra, an integrated network of neuronal and endocrine signals is coordinated by a conductor to synchronize the gut ecosystem. The gastrointestinal tract (GIT), often described as a “second brain,” is equipped with extensive neural circuits and hormone-secreting cells that ensure the temporal and spatial coordination of digestion, absorption, and microbial activity [56,57]. This orchestrated regulation is essential for maintaining digestive efficiency and metabolic homeostasis in the host. The enteric nervous system (ENS), an intricate network of over 100 million neurons embedded within the gut wall, is the heart of such orchestration [57].

So, digestion is regulated through a complex interplay of neural signals from both the enteric (ENS) and central nervous systems (CNS), with communication between gut regions helping coordinate digestive processes. Enteric neurons use diverse chemical messengers that act on numerous receptors, offering multiple potential targets for modifying digestive function. The ENS autonomously regulates key functions such as peristalsis, secretion of digestive juices, vascular tone, and communication with immune and epithelial cells. Importantly, ENS neurons are equipped with receptors that sense microbial metabolites, including SCFAs, bile acids, and tryptophan derivatives. These microbial signals modulate neuronal activity and, in turn, affect gut motility, secretion rates, and mucosal barrier function [58]. Figure 2 schematically demonstrates host–gut microbiota cooperation via SCFAs. This bidirectional communication enables a responsive and flexible digestive environment that adapts to both dietary and microbial fluctuations.

The influence of the microbiota on neuroendocrine regulation extends beyond simple feedback. Microbes are capable of producing a variety of neuroactive substances, including gamma-aminobutyric acid (GABA), dopamine, serotonin precursors, and acetylcholine-like compounds. These substances can affect the ENS directly or signal to the CNS via the bloodstream and vagus nerve. Moreover, microbial metabolites interact with the hypothalamic–pituitary–adrenal (HPA) axis, influencing systemic stress responses, appetite regulation, and metabolic rate [61]. Those collectively show that the gut microbiota is not the only participant in digestion but also acts as an active endocrine organ with the capacity to influence host behavior and physiology on a systemic level.

An additional layer of complexity is introduced by the nutrient-sensing mechanisms embedded in both the host and microbial components of the gut. Enteroendocrine cells are equipped with receptors for glucose, amino acids, fatty acids, and microbial metabolites, allowing them to integrate diverse signals and modulate hormonal output accordingly. Similarly, microbes adjust their gene expression profiles—including the expression of carbohydrate-active enzymes (CAZymes)—in response to available substrates and host-secreted molecules [14,15]. This coordinated sensing system enables the host and microbiota to anticipate and respond to dietary inputs in a synchronized manner, optimizing digestion and nutrient utilization.

Disruption of this finely tuned orchestration can have profound consequences. Factors such as antibiotic treatment, poor diet, chronic stress, aging, and disease can impair neuroendocrine signaling, disturb microbial composition, and reduce enzymatic efficiency [32,35]. Dysregulated hormonal responses may lead to impaired satiety signaling, delayed gastric emptying, or excessive bile acid accumulation, each of which can alter microbial niches and exacerbate dysbiosis [30,62]. Likewise, vagal dysfunction may impair feedback mechanisms essential for maintaining motility and mucosal health. The resulting breakdown in coordination can contribute to a wide range of disorders, from irritable bowel syndrome and metabolic syndrome to mood disorders and neurodegeneration [9,20].

The therapeutic implications of these insights are substantial. Targeting neuroendocrine regulators and their interaction with the microbiota opens new possibilities for improving digestive and systemic health. Prebiotics and dietary fibers that enhance SCFA production can stimulate beneficial hormone release and modulate ENS activity [7,37]. Probiotic strains selected for their ability to produce neuroactive compounds may serve as “psychobiotics” to improve mood and cognition. Personalized nutrition strategies that consider individual hormonal profiles, stress levels, and microbiota composition could optimize digestive efficiency and metabolic outcomes [62,63]. Furthermore, emerging technologies such as bioelectronic medicine and synthetic biology offer the potential to modulate the gut–brain axis in real time.

In conclusion, digestion is not merely a chemical process but a symphony conducted by an intricate network of neural and hormonal signals, finely attuned to the metabolic and structural contributions of the gut microbiota. Understanding this orchestration enriches our view of digestion from a linear sequence of breakdown steps to a dynamically regulated, co-evolved system. By embracing this perspective, future research and clinical practice can move toward integrative approaches that preserve and restore harmony in the digestive orchestra.

Moreover, the circadian regulation of both host and microbial processes adds another axis of orchestration. Circadian rhythms, driven by central and peripheral clocks, influence feeding behavior, hormone secretion, gut motility, and microbial activity. Disruption of these rhythms, as seen in shift work, jet lag, or irregular eating patterns, leads to desynchronization of digestive processes and microbial dysbiosis. For example, it was demonstrated that the intestinal microbiota, in both mice and humans, exhibited diurnal oscillations that are influenced by feeding rhythms, leading to time-specific compositional and functional profiles over the course of a day [38]. Interestingly, disruption of the host circadian clock via genetic ablation or jet lag led to microbiota dysbiosis driven by impaired feeding rhythms, resulting in glucose intolerance and obesity that are transferable to germ-free mice, thus revealing a microbiome-dependent mechanism linking circadian disruption to metabolic disorders in humans [38]. Restoration of rhythmicity may be a promising strategy to optimize host–gut microbiota cooperation and improve metabolic outcomes in such cases.

A systems biology perspective, integrating data from transcriptomics, metabolomics, and microbiota profiling, is essential to fully understand the complex orchestration of digestion. Computational models simulating gut neuroendocrine–microbiota interactions can help predict the impact of dietary or pharmacologic interventions on enzyme activity, microbial composition, and host health. Such models can also inform the development of synthetic microbial communities or engineered probiotics tailored to specific host needs. Moving forward, personalized interventions targeting the neuroendocrine pathways, either through diet, microbiota modulation, or behavioral change, offer a powerful approach to restore symphony in the gut ecosystem and mitigate digestive, metabolic, and neuropsychiatric disorders.

An intriguing question remains: Who is the conductor of the orchestra? This question arose many times, but it was not answered. Individual fast responses are coordinated by the nervous system, and slow responses are coordinated by the hormonal system. Together, they form the neurohumoral system. But that is not the whole story. The neurohumoral system does not operate in isolation. It cooperates tightly with other systems, including both its components of the digestive system, the host and microbiome. How do they coordinate their operation? In a classic way, via the system of forward and (mainly) feedback signals. So, there is no single conductor, and the orchestra is conducted by several cooperating conductors. That together can be called a “collective wisdom”.

## 8. Conclusions and Perspectives

In this review, we have highlighted the intricate relationships between gut microbiota, host digestive processes, and the regulation of the activities of enzymes under the influence of neural, hormonal, and nutritional cues. A growing body of evidence demonstrates that the gut microbiota not only metabolizes dietary components but also actively shapes host physiology through the production of signaling molecules, modulation of the intestinal barrier, and interactions with host immune and endocrine systems.

Key gastrointestinal hormones, including gastrin, cholecystokinin, and secretin, play pivotal roles in coordinating digestive enzyme secretion in response to both dietary and microbial cues. Furthermore, nutrient sensing mechanisms and the redox state of the gut environment critically affect microbial activity, enzyme expression, and barrier integrity. These dynamic interactions underscore the gut as a highly adaptive ecosystem that is affected by both intrinsic and extrinsic factors.

Despite significant progress, many aspects of gut microbiota–host communication remain poorly understood. Differences in microbial strain responses to oxidative stress, diet, and metabolic demands present both a challenge and an opportunity for deeper functional characterization. The emerging concept of hormesis in microbial responses to environmental stressors, including oxidative agents and toxicants, adds another layer of complexity that deserves further investigation.

Looking forward, future research should focus on the following areas:

Strain-specific responses—elucidating how different microbial strains within the same species respond to oxidative and nutritional stresses will enhance our understanding of gut resilience and adaptability.Integration of multi-omics approaches—combining metagenomics, transcriptomics, metabolomics, and proteomics will provide comprehensive insights into the dynamic interplay between microbes, enzymes, and host responses.Personalized nutrition and gut microbiota modulation—advancing microbiota-targeted interventions, including prebiotics, probiotics, and postbiotics, tailored to individual microbiome compositions, could significantly improve metabolic and gastrointestinal health. Such an approach may be a cornerstone for precision nutrition, leveraging high-throughput sequencing, metabolomics, and bioinformatics to tailor dietary recommendations based on individual genetic profiles, microbiome composition, lifestyle, and environmental factors, aiming to optimize nutrient intake, promote a healthy microbiome, and reduce disease risk.Therapeutic targeting of the gut barrier—strengthening intestinal barrier function through dietary, microbial, or pharmacological means offers a promising avenue to counteract systemic inflammation and chronic diseases linked to gut dysbiosis.

Altogether, an integrated, systems-level understanding of the gut ecosystem—bridging microbiology, enzymology, physiology, and nutrition—will be crucial for developing next-generation strategies to support human health and mitigate gastrointestinal and metabolic disorders.

## Figures and Tables

**Figure 1 biomolecules-15-01151-f001:**
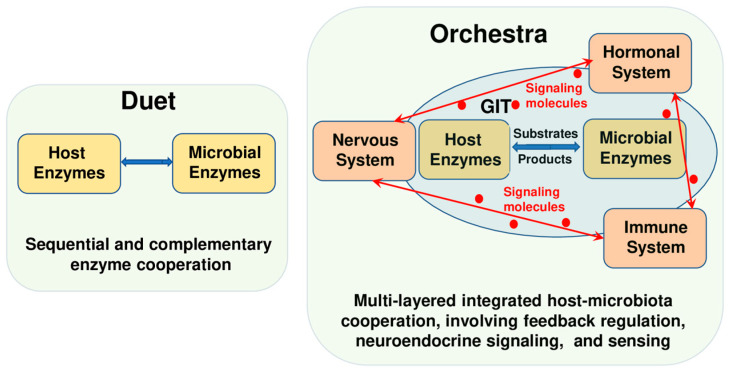
Conceptual model of enzymatic cooperation: “Duet” vs. “Orchestra”. Description in text. Abbreviations: GIT: gastrointestinal tract.

**Figure 2 biomolecules-15-01151-f002:**
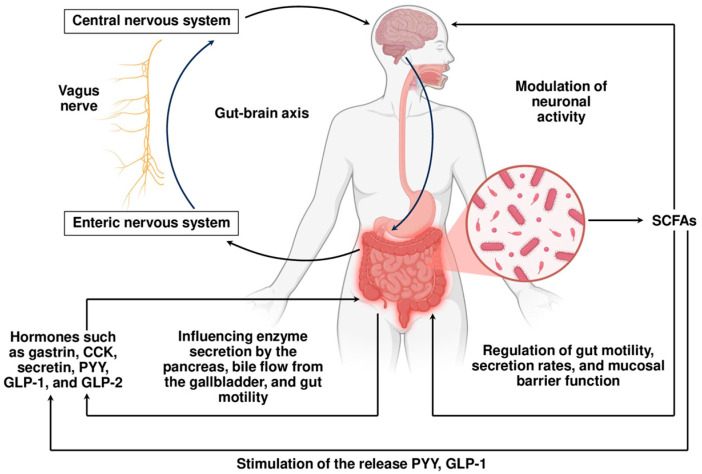
Orchestrating host–gut microbiota cooperation: plural feedback and forward signaling. The enteric and central nervous systems (ENS and CNS), connected via the vagus nerve, form a bidirectional communication system, the gut–brain axis, integrating signals from the CNS, ENS, and gut microbiota. SCFAs, such as acetate, propionate, and butyrate, are key microbial metabolites affecting gut physiology and influencing both the ENS and CNS via receptor-mediated signaling pathways, thereby modulating motility, barrier function, hormone release, and even behavior. In parallel, enteroendocrine hormones coordinate digestive functions, and enteric neurotransmitters convey gut-derived cues to the brain through vagal nerve signaling. This integrated network enables flexible adaptation to dietary and microbial fluctuations, while feedback loops maintain homeostasis in digestion, metabolism, and immune responses. Abbreviations: CCK—cholecystokinin; PYY—peptide YY; GLP-1/2—glucagon-like peptide 1/2; SCFAs—short-chain fatty acids. Figure partially created using BioRender.com (accessed on 25 July 2025). A range of gastrointestinal hormones, in parallel to the neural system, act as biochemical messengers coordinating the actions of digestive organs and relaying information to the CNS. Enteroendocrine cells, in response to luminal nutrients and microbial cues, secrete some hormones, and key among them are gastrin, cholecystokinin (CCK), secretin, peptide YY (PYY), and glucagon-like peptides (GLP-1 and GLP-2) [59,60]. These hormones influence enzyme secretion by the pancreas, bile flow from the gallbladder, and gut motility, while also modulating appetite and systemic glucose homeostasis. Of particular interest is the ability of microbial metabolites, especially SCFAs like butyrate and propionate, to stimulate the release of GLP-1 and PYY, thereby linking microbial fermentation to satiety and metabolic control [12]. The vagus nerve plays a critical role in connecting the gut and brain through the so-called gut–brain axis. Vagal afferent fibers transmit signals from the gut to the brain regarding nutrient content, hormone levels, and microbial metabolites, while efferent fibers regulate secretion, motility, and inflammatory responses [61]. The vagus nerve also mediates the effects of stress and emotional states on digestive physiology. Chronic stress, for example, can suppress vagal tone, reduce digestive secretions, and disrupt gut microbial composition, leading to a cascade of dysregulated processes. Conversely, interventions that enhance vagal activity, such as meditation, deep breathing, or electrical vagal stimulation, have been shown to improve gut function and microbial diversity.

**Table 1 biomolecules-15-01151-t001:** Comparison of host and microbial digestive enzymes by substrate and function.

Substrate	Host-Derived Enzymes	Microbial Enzymes	Primary Function
Starches and simple carbohydrates	Salivary amylase, pancreatic α-amylase	Amylases, pullulanases, maltases, isomaltases (CAZymes)	Hydrolysis of starch and oligosaccharides into monosaccharides
Non-digestible polysaccharides	None	Cellulases, hemicellulases, xylanases, pectinases	Fermentation of dietary fibers; SCFA production
Disaccharides	Sucrase, maltase, lactase (brush-border enzymes)	β-Galactosidase, β-fructofuranosidase	Breakdown of sucrose, maltose, and lactose into monosaccharides
Proteins and peptides	Pepsin, trypsin, chymotrypsin, peptidases	Microbial proteases, peptidases	Peptide hydrolysis; bioactive amine generation
Fats and lipids	Gastric and pancreatic lipases, phospholipases	Lipases, esterases	Hydrolysis of triglycerides and other lipids; production of microbial lipid metabolites
Bile acids	Liver secretion and reabsorption control	Bile salt hydrolases, hydroxysteroid dehydrogenases	Deconjugation and transformation of bile acids; host–microbiome signaling
Polyphenols and phytochemicals	Limited phase I/II metabolism (e.g., glucuronidation)	Glycosidases, esterases, decarboxylases	Microbial conversion into bioavailable phenolic compounds
Xenobiotics and drugs	Cytochrome P450s, conjugating enzymes (UGT, SULT, etc.)	Reductases, azoreductases, β-glucuronidases	Biotransformation, reactivation, or detoxification of xenobiotics

Abbreviations: CAZymes: carbohydrate-active enzymes; SCFAs: short-chain fatty acids; UGT: uridine 5′-diphospho-glucuronosyltransferase; SULT: sulfotransferase.

## Data Availability

No new data were created.

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
