# Peer review of "Symphony of Digestion: Coordinated Host–Microbiome Enzymatic Interplay in Gut Ecosystem"

_biomolecules, 2025, doi:10.3390/biom15081151_

Round 1
Reviewer 1 Report
Comments and Suggestions for Authors
The manuscript is elegantly written, concise and easy to read. The degree of novelty it brings is limited, since in general the process of human digestion has been very extensively studied, and there are standardized protocols worldwide that guarantee an adequate replication of what happens in our organism. However, there are some parts that I personally did not know about, such as the digestion of nucleic acids, which makes the manuscript much more interesting.
The resolution quality of the figures needs to be greatly improved, and there are some minor issues that need to be corrected:
Line 17: “SCFAs” are cited only once in the abstract. Thus, it is unnecessary to abbreviate it.
Figure 1 low quality, it is very difficult to read it. The same for Figure 2.
Line 240: delete a space after “(ENS)”.
Line 390: Conclusion section should not have references
Line 377-398: It is not advisable to underline in the main text
References are not in the font size recommended by the editor
Las two references lack in format such as bolds, italics, etc.
Reviewer 2 Report
Comments and Suggestions for Authors
The manuscript entitled “Symphony of Digestion: Coordinated Host-Microbiome Enzymatic Interplay in the Gut Ecosystem” is an interesting topic. However, there are still several parts that need to be revised:
- The paper introduces two models ("duet" and "orchestra") to describe host-microbiome enzymatic interactions. While creative, these metaphors could benefit from more precise definitions and clearer distinctions. For instance, the "orchestra" model mentions feedback loops but lacks detailed examples or mechanistic explanations. The authors should elaborate on how these models differ in practice and provide concrete evidence supporting their applicability.
- Section 6 highlights nucleic acid digestion as an overlooked area, but the discussion feels superficial. The authors briefly mention microbial nucleases and their roles but fail to connect this to broader themes like metabolic outcomes or disease implications (e.g., IBD, colorectal cancer). Expanding this section with specific microbial taxa, enzymatic pathways, and clinical relevance would strengthen the paper's novelty.
- Figures 1 and 2 are referenced but not described in sufficient detail within the text. For example, Figure 2's role in illustrating SCFA-mediated signaling is critical, yet the text only vaguely alludes to it. The authors should explicitly discuss key elements of each figure (e.g., arrows, labels) and their significance to the proposed models or mechanisms.
- Section 5 on therapeutic perspectives is broad, mentioning probiotics, postbiotics, and personalized nutrition without depth. The paper would benefit from concrete examples (e.g., specific probiotic strains, clinical trials) or challenges (e.g., scalability, individual variability). A table summarizing current interventions and their efficacy could enhance utility for readers.
- The brief mention of circadian regulation (Page 8) feels tacked on and underdeveloped. Given its potential impact on enzyme activity and microbial composition, this topic warrants a dedicated subsection or deeper exploration of how chrononutrition could be leveraged therapeutically, supported by recent literature.
- Terms like "dysbiosis" and "inflammaging" are used without consistent definition, which may confuse readers. For instance, "dysbiosis" is introduced in Section 4 but not clearly differentiated from general microbial imbalance. A glossary or explicit definitions upon first use would improve clarity, especially for interdisciplinary audiences.
Reviewer 3 Report
Comments and Suggestions for Authors
The author's idea is certainly not new, but this article presents it in the form of a clear composed review.
However, I have some comments and suggestions.
1. Figure 1, it would be more correct to present the ‘blow’ and ‘orchestra’ systems on identical metabolic pathways. In the picture and text presented, I understand that the systems do not apply to identical pathways. In this case it is necessary to stipulate this.
2. Part L214-225 - this part is not informative at all and is not relevant to the topic of the review!
3. L332-334 reference to a review article, although there should be a reference to a study.
Great remark - the article mostly cites review papers, whereas review articles should be based on experimental evidence. In this case we have a review article on reviews. I think that the authors should correct this point and expand the list of references with research articles.
The article is interesting to read, although it does not reveal a revolutionary approach. Competent physiologists, microbiologists clearly understand the complexity of the digestive system and the complexity of its regulation at the neurohumoral and microbial level.
Round 2
Reviewer 3 Report
Comments and Suggestions for Authors
The author has taken into account all my comments